


# Drought propagation and construction of a comprehensive drought index based on the SWAT-$K_{C'}$: A case study for the Jinta River basin in Northwestern China

Zheng Liang[1,2], Xiaoling Su[1,2], Kai Feng[1,2]

[1] College of Water Resources and Architectural Engineering, Northwest A & F University, Yangling 712100, China;
[2] Key Laboratory for Agricultural Soil and Water Engineering in Arid Area of Ministry of Education, Northwest A & F University, Yangling 712100, China

*Correspondence to*: Xiaoling Su (xiaolingsu@nwafu.edu.cn)

**Abstract.** Reliable drought monitoring and mastering the laws of drought propagation are the basis for regional drought prevention and resistance. Multivariate drought indicators considering meteorological, agricultural, and hydrological information may fully describe drought information; however, too short or missing hydrological variables in cold and arid regions make it difficult to monitor drought. This paper proposes a method combining SWAT and empirical Kendal distribution function ($K_{C'}$) for drought monitoring. The SWAT model, based on the principle of runoff formation, was used to simulate the hydrological variables of the drought evolution process. Three univariate drought indexes, namely meteorological drought (SPEI), agricultural drought (SSI), and hydrological drought (SDI) were constructed using parametric and non-parametric methods to analyze the propagation time of meteorological drought to agricultural drought and hydrological drought. The $K_{C'}$ was used to build a multivariable comprehensive Meteorology-Agriculture-Hydrology Drought Index (MAHDI) that takes into account meteorological, agricultural and hydrological drought to analyze the characteristics of a comprehensive drought evolution. The Jinta River in the inland basin of northwest China was used as the study area. The results show that agricultural and hydrological drought have a seasonal lag time for meteorological drought. The degree of drought in the river basin is high in the northern and low in the southern regions. The MAHDI captured drought conditions characterized by a univariate drought index; however, the ability to characterize mild and moderate droughts is stronger than severe droughts. The index also captured the occurrence and end of drought time; therefore, it is an acceptable comprehensive drought index. In addition, the comprehensive drought conditions showed insignificant drought trends in spring and summer, and showed insignificant warm and humidification trends in autumn, winter and annual scale. The results provided theoretical support for the drought control in the Jinta River Basin. This method may be applied for drought monitoring in other watersheds with a shortage of measured data.





# 1 Introduction

According to the fifth evaluation report of the Intergovernmental Panel on Climate Change (IPCC), climate change
characterized by temperature rise is the main concern of the global change in the past half-century with the most rapid
warming in the mid-latitude of the northern hemisphere. The arid inland river basins of China are mainly located in the
hinterland of the Eurasian continent in the mid-latitudes and are very sensitive to global climate change. Therefore, it is
particularly considerable to study the drought conditions of the inland river basins of China under the prevailing climate
change scenario.

Drought is a dynamic creeping phenomenon (Oikonomou et al., 2019; Ahmadi and Moradkhani, 2019); however, there is no
precise definition for the differences in hydro-meteorological variables and socio-economic elements (Mishra and Singh,
2010). Generally, the droughts are divided into four categories (Heim, 2002): (i) meteorological drought, referring to a
period of time with a lack of precipitation (Mishra and Singh, 2010; Dai, 2011). (ii) agricultural drought, referring to a period
when soil water is lower than the normal level (Dai, 2011) hindering crop growth and decrease in grain yield (Crow et al.,
2012; Panu and Sharma, 2002). (iii) hydrological drought, referring to a deficit in streamflow or groundwater resources
(Cammalleri et al., 2016). (iv) socio-economic drought, referring to a phenomenon that affects production, consumption, and
other socio-economic activities because of water shortage. There is a close relationship among the various droughts.
Insufficient precipitation for a long time leads to meteorological drought. When this situation lasts for a long period of time,
the soil water content decreases, which leads to a reduction in crop yield, resulting in agricultural drought. Insufficient
precipitation for a long period of time also causes a significant drop in surface water and groundwater, resulting in
hydrological drought. When all these three types of drought adversely affect social production and economic development,
socio-economic drought occurs.

Drought index is an important role in representing, measuring, and comparing the degree of drought for monitoring,
evaluating, and studying the development of drought (drought analysis). For example, the Standardized Precipitation Index
(SPI) (McKee et al., 1993) and the Standardized Precipitation Evapotranspiration Index (SPEI) (Vicente-Serrano et al., 2010)
are commonly used as meteorological drought indexes (Vicente-Serrano et al., 2012). The hydrological drought indices are
usually generated using streamflow, such as the Streamflow Drought Index (Nalbantis and Tsakiris, 2008) and the
Standardized Runoff Index (Shukla and Wood, 2008). The soil water content is the main variable to calculate the agricultural
drought index, for instance, the Crop Water Stress Index (Jackson et al., 1988) and the Standardized Soil Moisture Index
(Mishra et al., 2015). Calculation of the drought index requires a long time series of drought variables. However, the scarcity
of measured data is a major problem in the process of drought index construction. Therefore, to derive hydrological datasets,
indirect means have been attempted, such as watershed hydrological modeling. A recent study carried out by Dash et al.
(2019) applied the SWAT hydrological model to simulate the soil moisture data and develop Soil Moisture Stress Index (SSI)
for agricultural drought analysis. Further, Zhang et al. (2017) adopted a Variable Infiltration Capacity (VIC) hydrological
model to monitor soil moisture drought and construct a seasonal forecasting framework subsequently.



The shortcomings of a univariate drought index gradually emerged with the advancement of drought index research. As drought characteristics are usually interrelated, it is difficult for traditional drought studies that are based on univariate frequency analysis to reflect the complex and extensive characteristics of drought affecting social life. Therefore, it is necessary to develop a comprehensive drought index that integrates multiple variables related to drought. Keyantash and

Dracup (2004) used principal component analysis (PCA) to extract dominant drought variables to develop an Aggregate Drought Index (ADI) for comprehensive drought features (Rajsekhar et al., 2015). However, this method is a linear combination of related variables and could not reveal its non-linear structural characteristics. The Copula function can connect different marginal distributions, not only ensuring the independence of variables but also considering the correlation between them. It is one of the most commonly used connection methods at present and has been widely used in the field of

hydrometeorology. It is used to develop comprehensive drought indexes (Hao and AghaKouchak, 2013), for example, the Joint Drought Index (JDI) is constructed by copula using joint accumulated distribution of the runoff and precipitation (Kao and Govindaraju, 2010). Guo et al. (2019) used copula to propose an Improved Multivariate Standardized Reliability and Resilience Index (IMARRI) to fully appraise the dynamic risk of socio-economic drought. Wang et al. (2020) constructed a Standardized Precipitation Evapotranspiration Streamflow Index (SPESI) based on copula to comprehensively display

characteristics of meteorological and hydrological drought. Nevertheless, the limitation of Copula function to connect many variables is reflected in three or more dimensional drought indicators (Hao and Singh, 2013; Kao and Govindaraju, 2008,2010); this phenomenon is generally called "curse of dimensionality" (Hao and Singh, 2013). To overcome this limitation, this study applied empirical Kendall distribution function ($K_{C'}$) to construct a new comprehensive drought indicator by combining precipitation, evapotranspiration, soil water, and streamflow. The $K_{C'}$ is obtained by Nelson, which is

based on the generation function of the Archimedean copula function family (Nelsen, 2006). It is a probability integral transformation method and can transform multidimensional variable information into a single-dimensional (Hao et al., 2017). The Jinta River basin (JRB) is a tributary basin of the Shiyang River Basin (SRB) located in the mid-latitudes of Eurasia and is sensitive to global climate change (Wei et al., 2020). Therefore, it is important to study further drought conditions in the basin under the influence of climate change. The description of drought conditions is based on the construction of a drought

index, which is limited by the shortage of measured data. In addition, the construction of a comprehensive drought index is to be considered to reflect the drought situation comprehensively. However, the limitations of copula function in the construction of a multidimensional drought index outweigh the advantages of the copula. In this paper, the univariate drought indexes (SDI, SSI, and SPEI) established by measuring precipitation streamflow, and simulated soil water and evapotranspiration by the Soil and Water Assessment Tool (SWAT), respectively (Arnold et al., 1998; Zhang et al., 2010),

explored the lag time from hydrological and agricultural drought to meteorological drought. A new Meteorology-Agriculture-Hydrology Drought Index (MAHDI) was developed using the empirical Kendall distribution function based on the differences between precipitation and evapotranspiration, streamflow, and soil water. The MAHDI was also used to estimate the spatial distribution of the temporal tendency in different seasons. The specific objectives are to (i) to investigate the propagation time characteristics in the transition from meteorological to hydrological and agricultural drought; (ii) to

validate the accuracy of the MAHDI by comparing with univariate drought indexes (SDI, SSI, and SPEI); (iii) to estimate
the spatial distribution of MAHDI's temporal tendency in different seasons.

## 2 Materials and methods

### 2.1 Study area

The Jinta River Basin (JRB) with an area of 841 km2 originate from the Qilian Mountain and is a tributary of the Shiyang
River basin located in Gansu Province, China (see Fig. 1). It is bounded between the 100°57′E and 104°12′E longitudes and
the 37°02′N and 39°17′N latitudes. The JRB is located in the middle of the eight sub-basins, adjacent to the Zamu River in
the east and to the Xiying River in the west, as shown in Fig. 1(a). The terrain of the JRB is higher in the south and lower in
the north, sloping from southwest to northeast. The altitude ranges from 1890 meters to 4780 meters, with an average
altitude of 3000 m (Fig. 1(b)). The annual precipitation in the basin ranges from 200 mm to 500 mm, and the annual
evaporation is about 700–1200 mm.

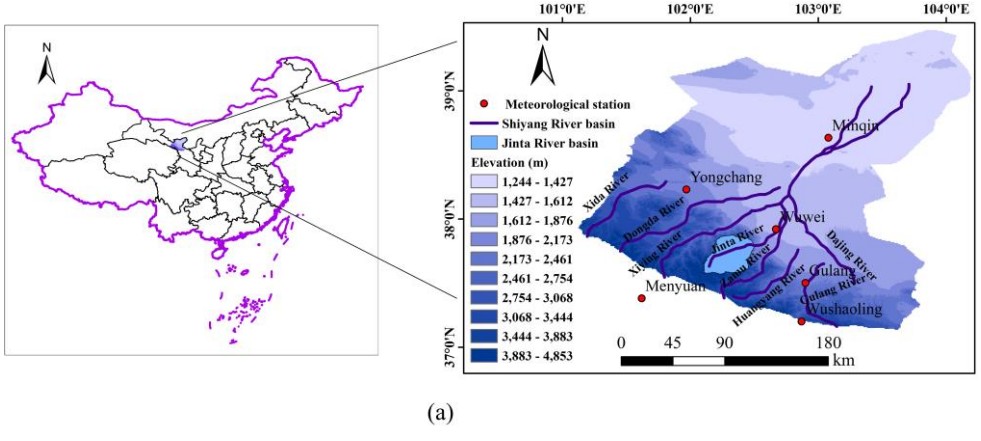

(a)

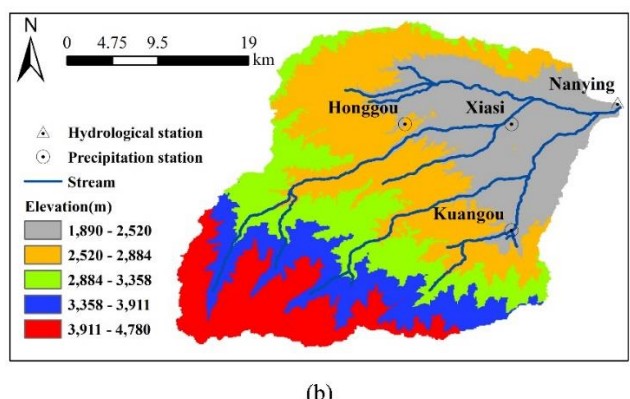

(b)

**Figure 1: Basic information about the Jinta River Basin. (a) The geographic location of the Jinta River in northwestern China; (b) Precipitation and hydrological stations in the Jinta River basin.**



## 2.2 Data sources

Digital Elevation Model (DEM) with a spatial resolution of 30 m provided by the Geospatial Data Cloud site, Computer Network Information Center, and Chinese Academy of Sciences (http://www.gscloud.cn) was used for watershed delineation. The digital soil map was obtained from the Harmonized World Soil Database (HWSD, version 1.1) developed by the Food and Agricultural Organization of the United Nations (FAO−UN). The map provided data for 5000 soil types containing two soil layers information (0−30cm and 30−100cm depth). The land-use data (30 m × 30 m) was derived from the satellite remote sensing image data of Landsat TM provided by the Geographical Information Monitoring Cloud Platform (http://www.dsac.cn/DataProduct). The observed climatic information of precipitation, maximum air temperature, minimum air temperature, wind speed, and relative humidity was obtained from six meteorological stations shown in Fig. 1(a) and three precipitation stations shown in Fig. 1(b). The monthly river discharge data for the model calibration and validation were obtained from the Nanying hydrological station at the Hydrology and Water Resources Bureau of Gansu Province for the period of 1984−2012.

## 2.3 SWAT model

The SWAT model developed by the Agricultural Research Service of the United States Department of Agriculture (USDA-ARS) is a continuous-time, semi-distributed, and physics-based water quality model (Arnold et al., 1998; Gassman et al., 2014; Romagnoli et al., 2017) for simulating hydrological cycle, plant growth cycle and transportation of sediments (Arnold et al., 1998; Pyo et al., 2019; Stefanidis et al., 2018; Wu et al., 2011). The SWAT model delineates a catchment into sub-basins based on the stream network and topography and subsequently into Hydrological Response Units (HRUs) representing different combinations of soil types, land use, and slope. The simulation calculation of soil effective moisture content, surface runoff, nitrogen content, and sediment yield are carried out, respectively for each of the HRUs. The hydrological part of the model is based on the water balance equation in the soil profile with processes, including precipitation, surface runoff, infiltration, evapotranspiration, lateral flow, percolation, and groundwater flow (Arnold et al., 1998; Kiniry et al., 2005).

$$SW_t = SW_0 + \sum_{i=1}^{t}\left(R_{day,i} - Q_{surf,i} - E_{\alpha,i} - w_{seep,i} - Q_{gw,i}\right), \tag{1}$$

where $SW_t$ is the final soil water content at time period $t$ (mm); $SW_0$ is the initial soil water content (mm); $t$ is the time (no. of days); $R_{days,i}$ is the amount of rainfall on $i$ th day(mm); $Q_{surf,i}$ is the amount of surface runoff on $i$ th day (mm); $E_{\alpha,i}$ is the amount of evapotranspiration on $i$ th day (mm); $w_{seep,i}$ is the amount of water giving recharge to groundwater from the soil profile on $i$ th day (mm); $Q_{gw,i}$ is the amount of return flow on $i$ th day (mm).

The runoff simulation of the watershed mainly consists of evapotranspiration, surface runoff, soil water and groundwater. The SWAT model has two main methods for estimating surface runoff, which are predicted by the Soil Conservation Service (SCS) Curve (Bouraoui et al., 2005). The channel routing uses the Muskingum method or the variable storage coefficient





model, including the migration of water, sediment, nutrients, and pesticides in the river network. Simultaneous calculation of reservoir confluence is also required. Evapotranspiration in the SWAT model refers to the process of surface water transforming into water vapor, including water evaporation, transpiration, and sublimation retained by tree crown as well as soil water evaporation. A part of the soil water is absorbed by plants or lost by transpiration, a part of it supplies the
groundwater, and the other part forms runoff on the surface. Groundwater runoff exists in the form of base flow calculated by groundwater storage and continuous runoff in the dry season.

## 2.4 Univariate drought index

A drought index contains a clear physical mechanism (Keyantash and Dracup, 2002) and is the main tool for quantitative analysis of drought characteristics. In addition, it can monitor the situation of the start time, the end time, duration, intensity,
and spatial range of drought. Therefore, the construction of the drought index is the basis of drought research. The difference of precipitation and evapotranspiration (P-ET), soil moisture (SM) and streamflow (D) simulated by the SWAT model were used to construct meteorological drought index (SPEI), agricultural drought index(SSI), and hydrological drought index (SDI) for four time steps (1,3,6 and 12 months) using parametric and non-parametric methods.

### 2.4.1 Parametric methods

The monthly sequence for each drought variable was fitted one by one by selecting an appropriate distribution function. The maximum likelihood method was applied to estimate the relevant parameters of the distribution function, the K-S test was used to test the fitting priority, and the Akaike's Information Criterion (AIC) was used to select the optimal fitting function. The cumulative probability distribution for each drought variable was then transformed into the standard normal distribution. Finally, the inverse function of the normal distribution was used to calculate the drought index.
i) The distributions selected in this study include Gamma distribution, Log-Normal distribution, Weibull distribution, Normal distribution, and Logistic distribution. Assuming that each distribution was suitable for the related drought variable series of each time scale, the maximum likelihood method was used to fit the parameter estimation. For a probability density function $f(x, \theta)$, $\theta$ is the parameter to be estimated and $X_1, X_2, X_3, \ldots, X_n$ is a sample from the population. If $x_1, x_2, x_3, \ldots, x_n$ is the sample value, the steps of the most valuable method are as follows:
a)  Construct the likelihood function of the sample:

$$L(\theta) = L(x_1, x_2, \ldots, x_n; \theta) = \prod_{i=1}^{n} f(x_i, \theta) , \qquad (2)$$

b)  The Log-likelihood function is given as:

$$lnL(\theta) = \sum_{i=1}^{n} f(x_i, \theta) , \qquad (3)$$

c)  Take the derivative of the parameter $\theta$ in Eq. (3) and make the derivative value 0:





$\frac{dlnL(\theta)}{d\theta} = 0$ ,                                                                                      (4)

d)  Solve the likelihood equation to get the maximum likelihood estimate $\hat{\theta}$ for the parameter $\theta$.

ii) The Kolmogorov-Smirnov(K-S) test is suitable for testing the goodness-of-fit of a dataset for most of the probability distributions regardless of the sample size by comparing the cumulative sample distribution with the cumulative distribution function specified by the null hypothesis. If the absolute value of the difference is within the specified range, the sample
obeys the assumed theoretical distribution. We assumed $H_0$ as the sample obeying the theoretical distribution, $H_1$ indicated that the sample did not follow the theoretical distribution. The statistic $D$ is constructed as:

$D = max|F_1(x) - F_2(x)|$ ,                                                                                  (5)

where $F_1(x)$ represents the cumulative distribution of samples, and $F_2(x)$ represents the theoretical distribution. At the selected significance level of α (α = 0.05), if $D > D(n, \alpha)$ ($n$ is the sample size), the $H_0$ is rejected, and $H_1$ is accepted;
otherwise, $H_0$ is accepted and $H_1$ is rejected.

iii) Akaike's information criterion (AIC) is a standard to measure the goodness of the statistical model fitting founded and developed by Japanese statistician Akaike. It weighs the complexity of the estimated model and the goodness of the model fitting data and is given as:

$AIC = 2m - lnL$,                                                                                              (6)

where $m$ is the number of parameters estimated by the distribution function, $L$ is the maximum likelihood function value. As increasing the number of free parameters improves the goodness of fitting, AIC encourages the goodness of data fitting but tries to avoid over fitting. Therefore, the priority model should be the one with the lowest AIC value. A lower AIC value indicates a better fit.

According to the AIC, the optimal theoretical distribution was selected. The inverse standardized value of the theoretical
distribution value corresponding to each drought variable was taken as the parametric drought index:

$DI_p = \emptyset^{-1}(P)$,                                                                                       (7)

where $DI_p$ is the non-parametric drought index value, $\emptyset$ is the standard normal distribution function, and $P$ is the theoretical cumulative probability.

**2.4.2 Non-parametric method**

If the four theoretical distributions for a certain drought variable could not pass the K-S test in the process of building a parametric drought index, the non-parametric method was used to build the drought index.

$P(x_i) = \frac{i-0.44}{n+0.12}$,                                                                               (8)





where $n$ is the length of the sequence, and $i$ is the order when the sequence of variables is ascending. The inverse standardization of the empirical cumulative probability is the non-parametric drought index expressed as:

$$DI_{nonp} = \emptyset^{-1}(P), \qquad (9)$$

where $DI_{nonp}$ is the non-parametric drought index value and $P$ is the empirical cumulative probability.

## 2.5 Trivariate drought index

The Kendall distribution function is obtained by Nelson according to the generation function of the Archimedean copula function family. It is a probability integral transformation method (Nelsen, 2006) and can transform multidimensional variable information into single-dimensional variable information. As some copula functions may not have the analytic expression of Kendall distribution function, this study used a non-parametric method to construct the empirical Kendall distribution function (Nelsen et al., 2003; Hao et al., 2017) expressed as:

$$K_{C'} = P = \frac{n_2}{n}, \qquad (10)$$

where $n_2$ is the number of sample satisfying $C'(i/n, j/n, k/n) \leq p$ ($C'$ is the empirical Copula function) and $n$ is the total number of samples. The express of empirical copula is given as (Hao et al., 2017):

$$C'\left(\frac{i}{n}, \frac{j}{n}, \frac{k}{n}\right) = P = \frac{n_1}{n}, \qquad (11)$$

where $n_1$ is the number of the samples $(x_m, y_m, z_m)$ satisfying $\left(x_m \leq x_{(i)} \ and \ y_m \leq y_{(j)} \ and \ z_m \leq z_{(k)}\right)$ and $1 \leq m \leq n$.

The empirical Kendall distribution function was used to join the three drought-related variables to obtain a trivariate drought indicator by inverse standardization:

$$MAHDI = \emptyset^{-1}(P), \qquad (12)$$

where MAHDI is the trivariate drought index value and $P$ is the three-dimensional cumulative probability.

## 3 Results and discussion

### 3.1 SWAT model calibration and validation

In order to calibrate and validate the runoff related parameters, we applied the SWAT Calibration and Uncertainty Programs (SWAT-CUPs). The calibration period was taken as 1986−2000, and the validation period was taken as 2001−2012. In addition, a warm-up period of 1984−1986 was considered to minimize the uncertainty caused by the initial environment of the model (Zhang et al., 2019). In the SWAT-CUPs, the Sequential Uncertainty Fitting Version 2(SUFI-2) algorithm (Abbaspour et al., 2007) was chosen for parameter sensitivity and model uncertainty analysis (Abbaspour et al., 2015).





**Table 1: Sensitivity analysis and final value range of parameters in Jinta River basin**

| Parameters | Meaning of parameter | Initial range | Fitted value | Method | $t$-Stat | $p$-Value |
|---|---|---|---|---|---|---|
| CN2* | SCS runoff curve number for moisture condition II | 35−98 | 70.51−95.03 | Replace | 43.10 | <0.01 |
| SOL_AWC | Available water capacity of the soil layer | 0−1 | 0.008 | Replace | 38.69 | <0.01 |
| TLAPS | Temperature lapse rate | -10−10 | 2.83 | Replace | -7.72 | <0.01 |
| SOL_K | Saturated hydraulic conductivity | 0−2000 | 99.65 | Replace | 5.04 | <0.01 |

Note: As the CN2 of different land-use types is calibrated separately, a range of the optimal CN2 values is provided

Table 1 shows the top four sensitive parameters of the JRB and their initial and fitted values. The CN2, the comprehensive response of the underlying surface characteristics, was the most sensitive parameter in the hydrological process. The value of CN2 was calibrated to 70.51−95.03 for different land use types. In order, the next sensitivity parameters were SOL_AWC,

TLAPS, and SOL_K. Among them, SOL_AWC and SOL_K are soil-related sensitivity parameters, and their fitted values were 0.008 and 99.65, respectively. The TLAPS is a parameter related to temperature, and its optimal value was 2.83.

**Table 2: Model performance in monthly streamflow in JRB**

| Indicators | $P$-factor | $R$-factor | $R^2$ | $E_{NS}$ |
|---|---|---|---|---|
| Calibration period (1986~2000) | 0.72 | 0.65 | 0.76 | 0.75 |
| Validation period (2001~2012) | / | / | 0.73 | 0.72 |

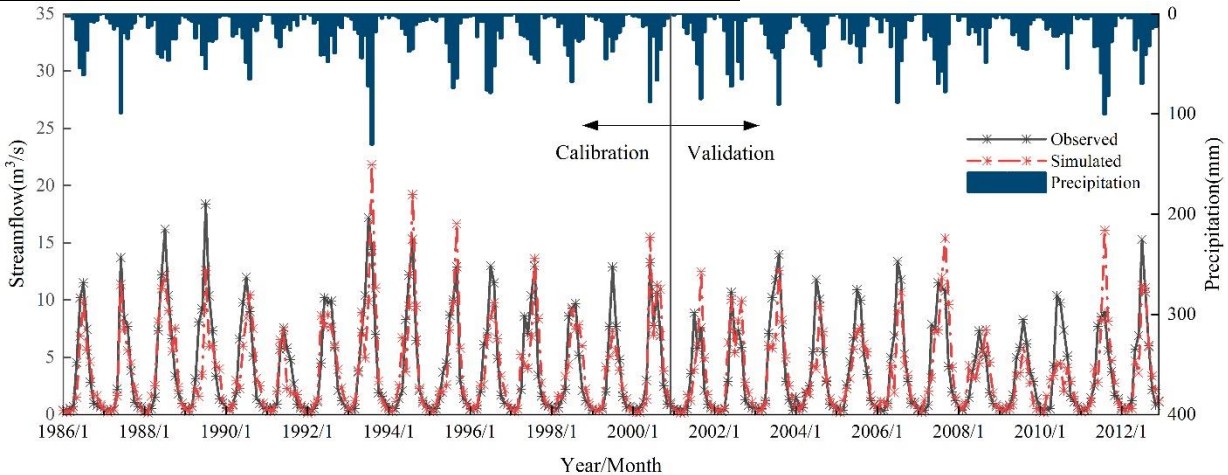

**Figure 2: Simulated and observed monthly streamflow series relative to precipitation (P) during the calibration and validation**
**periods in the JRB**

Uncertainty of the model was adjudged on the basis of $P$-factor and $R$-factor indicators (Abbaspour et al., 2007). When the $P$-factor > 0.7 and $R$-factor < 1.5, the uncertainty of the model was considered as acceptable, and the parameter ranges were





taken as the calibrated parameters. Table 2 shows that two indicators are in the acceptable range, with a $P$-factor of 0.72 and

an $R$-factor of 0.65. To measure the model performance, we selected the coefficient of determination ($R^2$) and the Nash-Sutcliffe simulation efficiency ($E_{NS}$) (Table 2). The simulation results showed that the $R^2$ and $E_{NS}$ in the JRB were 0.76 and 0.75, respectively, for the calibration period and 0.73 and 0.72, respectively, for the validation period. Fig. 2 shows the plots for the simulated monthly streamflow against the observations. The figure indicates that the simulated and observed monthly streamflow were in good agreement with the period considered, and their changes followed the P pattern. Overall, the model performance was satisfactory for subsequent analysis.

## 3.2 Drought characterization

The SPEI, SSI, SDI, and MAHDI were applied to all sub-basins at 1,3 and 12-month time scales. For calculating the four indexes, precipitation, evapotranspiration, soil moisture, and streamflow data from 1986 to 2012 were adopted in this study. Thresholds of the indexes were divided according to the SPI (McKee et al., 1993).

### 3.2.1 The propagation time from meteorological drought to hydrological and agricultural drought

To study the lag-time from meteorological to hydrological drought, the relationships between the SDI and the SPEI with various time scales were explored. Similarly, the relationship between SSI and SPEI on different time scales also reflects the response propagation time of agricultural drought to meteorological drought. We can mention No. 6 sub-basin where the hydrological station is located as an example.

Figure 3 shows the correlation coefficients between monthly SDI and SPEI with various time scales at the No.6 sub-basin.

High correlation coefficients (>0.7) of SDI and SPEI were observed for spring and summer with the time scales from 2 to nine months. Low correlation coefficients (>0.4) were observed for autumn and winter with about 6~9 months scale. The lag time in spring and summer is more, as obvious, showing certain seasonal characteristics, whereas the same in autumn and winter have inconspicuous seasonal characteristics. A reasonable explanation about this phenomenon might be more sources of recharge (rainfall and iceberg snowmelt) in spring and summer, while groundwater was the only source of recharging the

river in autumn and winter, which is related to the water stored during spring and summer.

Similarly, Figure 4 depicts the propagation time of agricultural drought responding to meteorological drought at No.6 sub-basin, which also shows an obvious seasonal characteristic. In summer, the lag time approximately concentrated two months with a correlation coefficient value of higher than 0.8, while the response time in other seasons is longer. The lag time from agricultural drought to meteorological drought is, therefore, the shortest in summer. This may be the result of high soil

moisture due to high rainfall during the season. The propagation time in spring is three months longer than that in summer, which may be because of the potential influence of snowmelt. In the autumn and winter, there is longer lag time (6~12 months) for responding to the meteorological drought, possibly due to the infiltration of soil water during the preceding months.


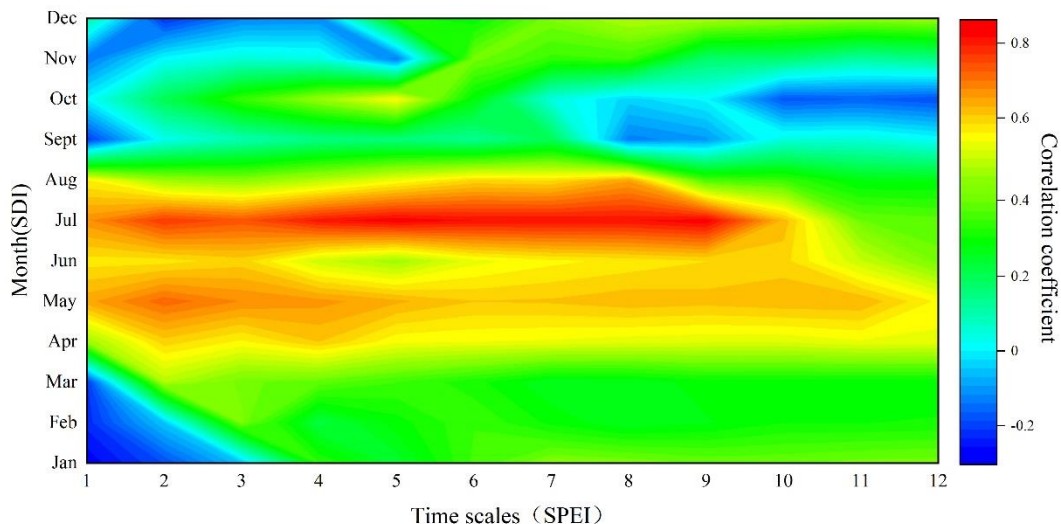

**Figure 3: The correlation coefficients between monthly SDI and SPEI with different time scales at the No.6 sub-basin**

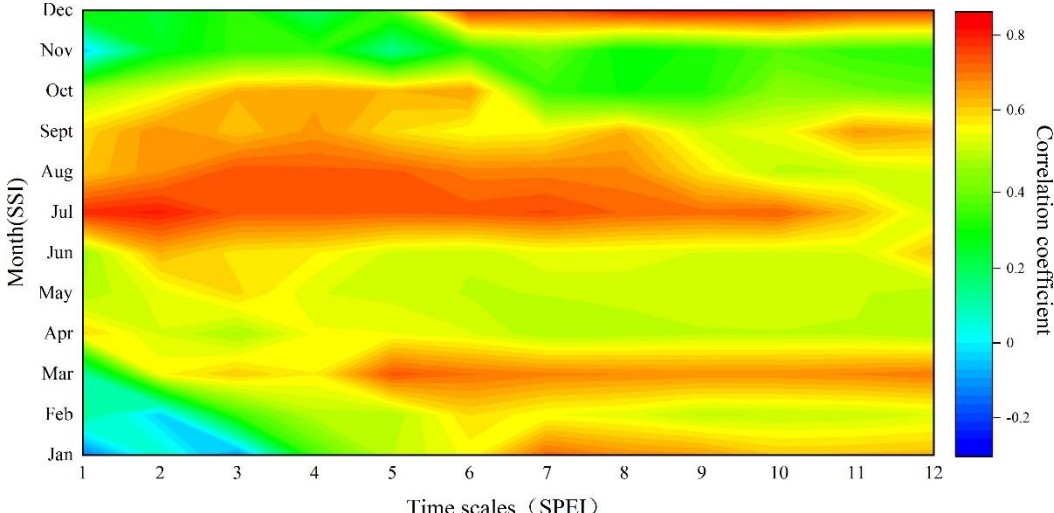

**Figure 4: The correlation coefficients between monthly SSI and SPEI with different time scales at the No.6 sub-basin**

As compared to Fig. 3, Figure 4 is more precise in showing that the lag time from agricultural drought to meteorological drought increases with a decrease in temperature and precipitation, and there is a clear gap between different seasons. However, the time by which the hydrological drought lags behind the meteorological drought is not obvious. The distribution of glaciers in the upper reaches of the Jinta River and the significantly longer time of soil water infiltration than that of confluence formation might make the lag time of agricultural drought to meteorological drought more obvious in

different seasons as compared to the lag time of hydrological drought to meteorological drought. Studying the lag time of





different types of droughts to meteorological droughts is helpful in predicting other droughts using meteorological droughts in the absence of measured data and may provide a theoretical basis for drought prevention.

### 3.2.2 SSI, SDI, SPEI and applicability analysis of MAHDI

To analyze the distribution of different droughts and the applicability of MAHDI, the year 1999 was selected for analysis.
The spatial distribution of SDI, SSI, SPEI, and MAHDI for the year 1999 is shown in Fig. 5. For SDI, severe drought was distributed in No.1, No.2, No.3, No.4, No.5, No.10, and No.18 sub-basins. Moderate drought was observed in No.6, No.7, No.8, and No.11 sub-basins, and mild drought was observed in the rest of the sub-basins. For SSI, extreme drought was distributed in No.1, No.2, and No.11 sub-basins, severe drought was located in No.3, No.5, No.8, No.9, No.17, No.18, and No.19 sub-basins, moderate drought was observed in No.4, No.6, No.7, No.10, No.14, No.16, No.21, and No.22 sub-basins,
and mild drought was distributed in No.12, No.13, No.15, No.20, and No.23 sub-basins. For SPEI, severe drought was observed in the No.18 sub-basin, moderate drought was located in No.1, No.2, No.3, No.4, No.5, No.7, No.8, No.9, No.10, and No.11 sub-basins, and mild drought was observed in the rest of the sub-basins. For MAHDI, severe drought was located in No.1, No.2, and No.18 sub-basins, mild drought was distributed in No.5, No.12, No.13, No.14, No.15, No.16, No.20, No.22, and No.23 sub-basins and moderate drought were located in the rest of the sub-basins. In conclusion, the drought in
the northern part of the basin is stronger than that in the southern part of the basin.

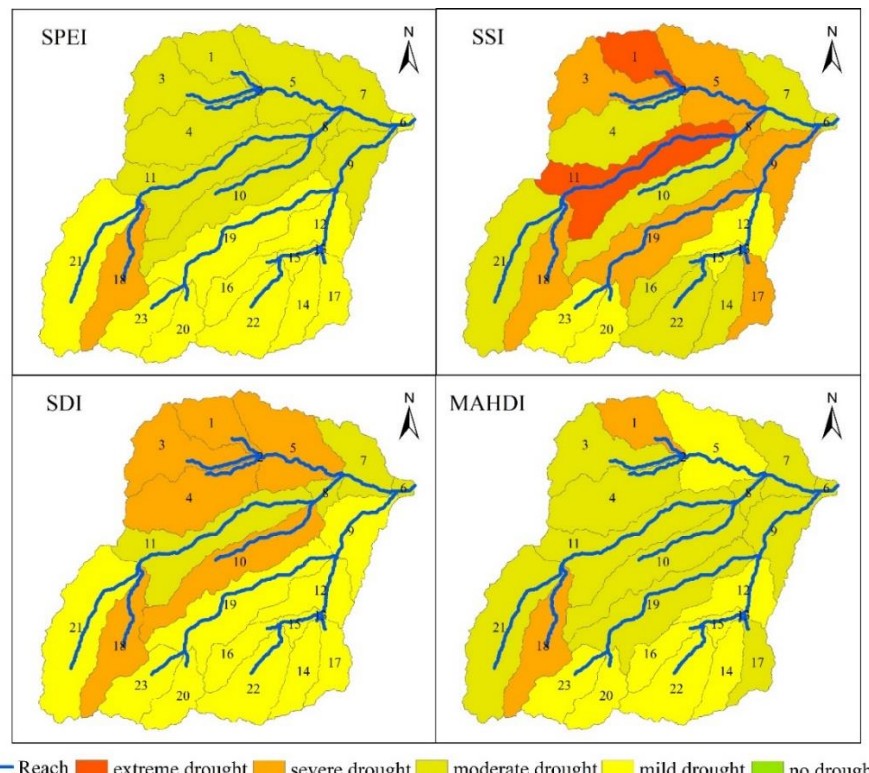

**Figure 5: Distribution of SDI, SSI, SPEI, and MAHDI at 12-month scale in JRB in 1999**





Different drought indexes showed different degrees of drought severity. For univariate drought indexes, SSI showed the highest degree of drought, followed by SDI, and SPEI showed the lowest degree of drought. The reason for SPEI to exhibit
the lowest degree of drought might be due to the warming and humidification of the Shiyang River Basin, which increased rainfall and temperature. As rainfall and temperature are the core elements of SPEI, the meteorological drought was alleviated (Guo et al., 2016). The highest degree of drought shown by SSI might be because of topographic factors. There are many glaciers in the JRB, and the river confluence speed is faster than the soil infiltration speed resulting in low soil water storage capacity. Besides, the calibrated value of SOL_AWC by the SWAT model was only 0.008(Table 1), showing that the
water storage in the soil layer in the basin was very small. The MAHDI captured all the mild and moderate droughts shown by SDI, SSI, and SPEI, as well as the severe drought in some sub-basins. However, it could not capture the extreme drought shown by SSI in the No.1 and No.11 sub-basins. Therefore, it may be concluded that the MAHDI's ability to capture mild and moderate droughts is stronger as compared to its ability to capture severe and extreme droughts. This might be due to the lack of empirical Kendall function's ability to deal with extreme values.

**Table 3: The capture time of various drought indexes for the drought event that occurred in the period 1991–92 and 1999–2000 at No.6 and No.8 sub-basins**

|  | Reach 6 |  |  | Reach 8 |  |  |
| --- | --- | --- | --- | --- | --- | --- |
|  | Index | Onset time | End time | Index | Onset time | End time |
|  | SPEI | 06/1991 | 09/1991 | SPEI | 06/1991 | 09/1991 |
| 1991–92 | SSI | 07/1991 | 01/1992 | SSI | 06/1991 | 01/1992 |
|  | SDI | 07/1991 | 01/1992 | SDI | 07/1991 | 01/1992 |
|  | MAHDI | 06/1991 | 01/1992 | MAHDI | 06/1991 | 01/1992 |
|  | Index | Onset time | End time | Index | Onset time | End time |
|  | SPEI | 06/1999 | 12/1999 | SPEI | 06/1999 | 12/1999 |
| 1999–2000 | SSI | 07/1999 | 05/2000 | SSI | 06/1999 | 05/2000 |
|  | SDI | 08/1999 | 05/2000 | SDI | 07/1999 | 05/2000 |
|  | MAHDI | 07/1999 | 05/2000 | MAHDI | 06/1999 | 05/2000 |

Drought events in the period of 1991–92 and 1999–2000 in No. 6 and No. 8 subbasins were selected to verify MAHDI's ability to capture the onset and end time of drought events (Table 3 shows the capture time of various drought indexes for
these drought events). Among the univariate drought indexes, the SPEI captured the drought earlier than any other indexes, and it also captured the earliest end time of drought. The starting and ending time of drought represented by SSI and SDI was later than that of SPEI, which made the drought to have a longer duration. This is because the rate of change in precipitation and temperature is the fastest, whereas runoff generation and soil water infiltration require a certain time, making the meteorological drought the most sensitive to environmental changes. Compared with univariate drought indexes,
the MAHDI characterized drought at the same time as that of the SPEI and was consistent with SSI and SDI's





characterization of the drought end time. It may be concluded that the MAHDI combined the advantages of SDI, SSI, and SPEI and is a comprehensive function of streamflow, soil water, precipitation, and temperature. Overall, the three-dimensional drought index MAHDI constructed in this paper is acceptable.

### 3.2.3 Drought temporal characterization

To assess the spatial characteristics of comprehensive drought temporal tendency in the JRB, we calculated the Man-Kendall (M-K) statistics. The M-K statistics with a significant level of MAHDI were represented for different seasons and a 12-month scale (Fig. 6). A positive M-K statistic indicates an increasing tendency of drought index and vice versa. Besides, the M-K statistic values also include a test of significance (significance level was $\alpha= 0.05$ and the threshold values were $\pm1.96$).

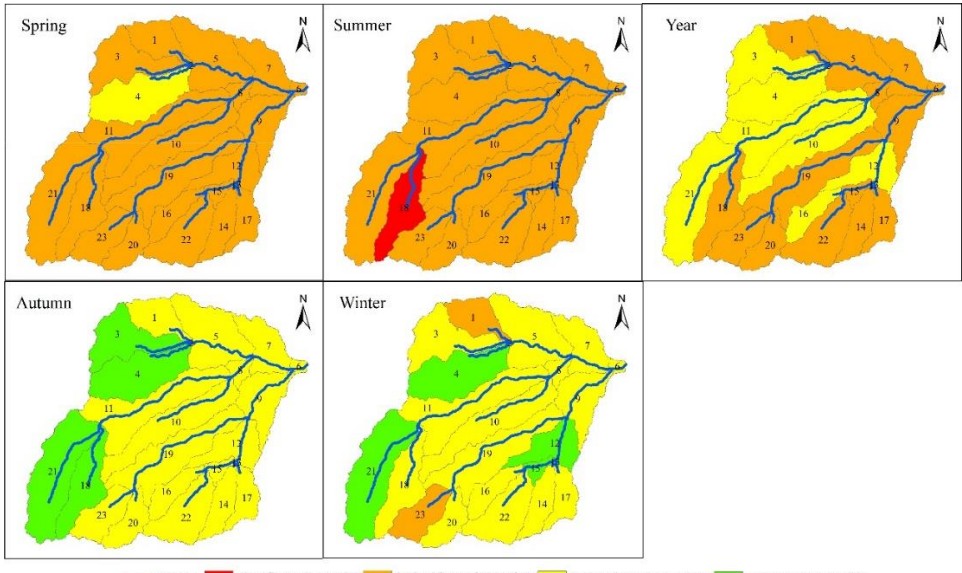

**Figure 6: M-K trend test of MAHDI in 3-month and 12-month scales in JRB**

Figure 6 shows different spatial characteristics of drought temporal trends for various seasons. In spring, the MAHDI of most of the sub-basins showed a non-significant decreasing trend and only No.4 sub-basin showed an insignificant increasing trend. In summer, the MAHDI for most of the sub-basins also showed an insignificant decreasing trend; No.18 sub-basin showed a significant decreasing trend. In autumn, the drought index in No. 3, No. 4, No. 21, and No. 18 sub-basins

showed a significant increasing trend, and the rest of the sub-basins showed an insignificant upward trend. In winter, the drought index in No.1 and No.23 sub-basins showed an insignificant decreasing trend and the changes in No. 4, No. 12, No. 13, No. 15, and No. 21sub-basins showed a significant increasing tendency; the rest of sub-basins showed an insignificant increasing trend. For a 12-month scale, the MAHDI's tendency mainly composed of an insignificant upward trend and an insignificant downward trend. The sub-basins with drought index showing an insignificant decreasing trend were No.3, No.4,

No.10, No.11, No.12, No.16, and No.21 and the rest of the sub-basins with drought index showed an insignificant increasing





trend. Therefore, it might be concluded that drought temporal trend has spatial differences and are influenced by seasonal characteristics and geographical conditions in the JRB during the study period.

Drought temporal tendency analysis can help people predict drought and take measures in advance to reduce the drought damages. Our results found that an insignificant decreasing trend of MAHDI mainly occurred in spring and summer, and

autumn and winter showed an insignificant increasing trend. About 1/3 of the sub-basins showed an insignificant decreasing trend and about 2/3 of the sub-basins showed an insignificant increasing trend for a 12-month scale. A possible explanation for this may be that global warming makes the climate in the upper reaches of the Shiyang River warmer and more humid (Guo et al., 2016). The trend of warming and humidification in autumn and winter is more obvious, which is consistent with the conclusions put forward by previous researchers (Zhou et al., 2012).

**4 Conclusion**

In this paper, the SWAT hydrological model is used as an indirect way to obtain hydrometeorological data and simulate the missing data to construct SDI, SSI, and SPEI at different time scales and analyzed the transfer relationship between different droughts. In addition, for the "dimensional disaster" phenomenon that occurs when the copula function is used to connect multidimensional variables, this study uses $K_{C'}$ to combine multiple hydrometeorological variables to construct a

comprehensive drought index MAHDI that can simultaneously characterize meteorological, agricultural, and hydrological drought, and analyzes the features of drought changes in the JRB. The following conclusions are derived from the research:

i) Agricultural and hydrological drought have certain lag to meteorological drought and the lag time has seasonal characteristics. The shortest lag time of about two months is observed in summer, followed by spring. The lag time in autumn and winter is the longest, mostly exceeding six months. The lag time of agricultural drought is more obvious than

that of hydrological drought, which may be because soil water infiltration time is longer than runoff concentration time.

ii) The degree of drought in the north of the basin is stronger than that in the south. The degree of agricultural drought is the strongest, followed by hydrological drought, and that of meteorological drought is the weakest. This is due to the low water storage capacity of the soil (the calibrated value of SOL_AWC by the SWAT model is only 0.008).

iii) MAHDI can capture the drought events that are captured by univariate drought indexes (SDI, SSI, and SPEI); however,

its description ability of mild and moderate drought is better than that of severe drought. In addition, it can timely capture the occurrence and end of drought events, which is a characteristic of an acceptable comprehensive drought index.

iv) MAHDI showed an insignificant downward trend in spring and summer and an insignificant upward trend in autumn and winter. For a 12-month scale, for about 1/3 of the sub-basins, the MAHDI showed an insignificant downward trend and for about 2/3 of the sub-basins it showed an insignificant upward trend. The drought situation in the JRB has been alleviated

under the influence of climate change.

The methods utilized in the present study to construct a comprehensive drought index (MAHDI) combining SWAT and Copula can be carried out in any other areas with deficient observed data. These results are emblematic of the drought





phenomenon in the JRB. However, the ability of MAHDI to characterize severe drought is relatively low, and further research is required to improve its ability to monitor severe drought.

**Author contribution**

S.X. contributed the central idea; L.Z. analysed most of the data, wrote the initial draft of the paper and finalized this paper; F.K. revised the manuscript.

**Competing interests**

The authors declare that there is no competing financial interest.

**Acknowledgements**

The authors are grateful for the support from the National Natural Science Fund in China (Grant Nos.51879222).

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
