# Peer review of "Drought propagation and construction of a comprehensive drought index based on the SWAT- $K_{C'}$ : A case study for the Jinta River basin in Northwestern China"

_Natural Hazards and Earth System Sciences, 2020_

## Referee Comment (RC1) · Anonymous Referee #1 · 4 Oct 2020

The authors proposed a multivariate drought index based on precipitation-temperature, soil moisture, and streamflow using the copula approach. These hydroclimatic variables are simulated by SWAT model. After the evaluation of drought propagation time, the comparison of the proposed drought index is evaluated based on the spatial extent, severity, and onset/end time of drought through comparisons with the individual indicators. Finally, the trend analysis of drought in the study area is evaluated based on the proposed index. My comments are as follows. Major comments: (1) Grammar and language: In several places, the grammar and language hinder understanding of

the meaning of statements (please see minor comments). This manuscript could be enhanced by carefully proofreading the context. (2) Structure: There are three main sections in the results. Are there any connections between them? For example, does the analysis of propagation time in 3.2.1 help the evaluation or explanation of 3.2.2? In the current form, it seems 3.2.1 and 3.2.2 are independent. This is also reflected in the title "propagation" and "construction". What is the connection?

(3) Novelty and evaluation: The novelty of the study seems to be the new drought index. A critical question following this is the evaluation of the index. How do we know that this proposed index is reliable? There is some comparison with the individual indicators. Apart from this, could you find some other evidence? For example, what about the public records or reports of historical drought events or losses in the study area?

Minor comments:

Line 31: please add related references on this. Lines 48-49: This statement can be more concise. Please also check the grammar. Lines 61-64: hard to understand. Please rewrite and make it clear. Line 68: "ensuring the independence of variables" what do you mean? Lines 70-71: Please check the grammar. Lines 82-84: This seems to be redundant, as similar statements have been shown in lines 33-34. Lines 86-87: "limitation...outweigh the advantages" . hard to understand. Please justify this or just remove it. Lines 251-253: "propagation time of agricultural drought to meteorological drought"? How so? (also in lines 264, 275, 279) Do you mean from meteorological drought to agricultural drought? These statements need revisions to make it clear. Line 267-268: Could you please elaborate how the "infiltration of soil water .." could induce the "long lag time"? Lines 299-301: The statement about "warming and humidification" leading to "increased rainfall and temperature" needs careful justification. Do you have references to support this?

Lines 38-309: "lack of empirical . . . with extreme values"? This limitation of the proposed index could be discussed in the conclusion or somewhere else.

---

## Referee Comment (RC2) · Anonymous Referee #2 · 25 Dec 2020

In this paper, author constructed single variable drought indexes such as SPEI, SDI, and SSI based on the SWAT hydrological model of Jinta River basin. The comprehensive drought index MAHDI based on empirical Kendall distribution function was developed, and the spatial and temporal trend characteristics of comprehensive drought in the study area were analyzed. This study is really interesting and helpful to drought monitoring and drought decision-making. Therefore, I recommend the manuscript to be minor revising after the following questions/issues addressed: 1. There are too many words in the abstract, so it needs to be simplified. 2. The sentence "Reliable

drought monitoring and mastering the laws of drought propagation are..." should be changed to "Monitoring drought and mastering the laws of drought propagation are...". 3. In the sentence "however, too short or missing hydrological variables in cold and arid regions make it difficult to monitor drought.", "too short hydrological variables" is unclear. 4. Drought control is mentioned in lines 48-49, but is not the subject of this manuscript. You should pay attention to the preciseness of the paper. 5. Line 70-71: please check the grammar. 6. Line 192: " DIp " should be the parametric drought index value, please check it. 7. Line 209-210: please add related references about "warming and humidification of the Shiyang River Basin". 8.Line 278-280: this sentence is not clear, please revise it.

---

## Author Comment (AC1) · 15 Jan 2021

Thank you very much for the reviewers' useful comments and suggestions on our manuscript. We have meticulously read your comments, and modified the manuscript accordingly. Detailed corrections are listed below point by point. Our responses to several comments are listed below:

Comment 1: There are too many words in the abstract, so it needs to be simplified. Response: Thanks for the reviewer's suggestion. As suggested by the reviewer, we have

[Figure]

revised the abstract and further simplified the language of the abstract. The specific changes are as follows: (1) Line 11, "drought information" were corrected as "drought conditions". (2) Line 12, "proposes" was corrected "proposed" and "Kendal" was corrected as "Kendall". (3) Line 14-17, "Three univariate drought indexes, namely meteorological drought (SPEI), agricultural drought (SSI), and hydrological drought (SDI) were constructed using parametric and non-parametric methods to analyze the propagation time of meteorological drought to agricultural drought and hydrological drought." were corrected as "Three univariate drought indexes, namely meteorological drought (SPEI), agricultural drought (SSI) and hydrological drought (SDI), were constructed using parametric or non-parametric method to analyse the propagation time from agricultural drought and hydrological drought to meteorological drought.". (4) Line 18, "takes" was corrected as "took" and "analyze" was corrected as "analyse". (5) Line 19-21, "The results show that agricultural and hydrological drought have a seasonal lag time for meteorological drought. The degree of drought in the river basin is high in the northern and low in the southern regions." were corrected as "The results showed that agricultural and hydrological drought had a seasonal lag time from meteorological drought. The degree of drought in this basin was high in the northern and low in the southern regions.". (6) Line 21-24, "The MAHDI captured drought conditions characterized by a univariate drought index; however, the ability to characterize mild and moderate droughts is stronger than severe droughts. The index also captured the occurrence and end of drought time; therefore, it is an acceptable comprehensive drought index." were corrected as "The MAHDI was proved to be acceptable for that it can catch drought conditions characterized by a univariate drought index and capture the occurrence and end of drought time. Nevertheless, its ability to characterize mild and moderate droughts was stronger than severe droughts.". (7) Line 24, "drought trends" were corrected as "aggravating trends". (8) Line 25, "warm and humidification" were corrected as "alleviating". (9) Line 26-27, "This method may be applied for drought monitoring in other watersheds with a shortage of measured data." were corrected as "This method provided the possibility for drought monitoring in other watersheds lacking measured

data.".

Comment 2: The sentence "Reliable drought monitoring and mastering the laws of drought propagation are . . ." should be changed to "Monitoring drought and mastering the laws of drought propagation are . . .". Response: Thanks to the reviewers for their helpful comments. According to the reviewer's comments, we modified this sentence. Line 9-10, "Reliable drought monitoring and mastering the laws of drought propagation are the basis for regional drought prevention and resistance." were corrected as "Monitoring drought and mastering the laws of drought propagation are the basis for regional drought prevention and resistance."

Comment 3: In the sentence "however, too short or missing hydrological variables in cold and arid regions make it difficult to monitor drought.", "too short hydrological variables" is unclear. Response: Thank you very much for your valuable comment. We have corrected this sentence as "However, too short or missing series of hydrological variables in cold and arid regions make it difficult to monitor drought.".

Comment 4: Drought control is mentioned in lines 48-49, but is not the subject of this manuscript. You should pay attention to the preciseness of the paper. Response: Thank you very much for your valuable comment. We changed "drought control" into "drought monitoring" in Line 26.

Comment 5: Line 70-71: please check the grammar. Response: Thank you very much for your valuable comment. We have deleted ". It is used to develop comprehensive drought indexes" in Line 69. And ", for example" were corrected as ". For example" in Line 70.

Comment 6: Line 192: "DIp" should be the parametric drought index value, please check it. Response: Thanks to the reviewers for your helpful comments. We have checked this "DIp" and corrected "non-parametric" in Line 189 as "parametric".

Comment 7: Line 209-210: please add related references about "warming and humid-

ification of the Shiyang River Basin". Response: Thanks for the reviewer's suggestion. As suggested by the reviewer, we found the location of "warming and humidification of the Shiyang River Basin", and we corrected "The reason for SPEI to exhibit the lowest degree of drought might be due to the warming and humidification of the Shiyang River Basin, which increased rainfall and temperature." in lines 305-307 as "The meteorological drought degree reflected by SPEI is the lowest, which is similar to that described by Thornthwaite aridity index (AI) constructed by Zhang et al. (2017) using rainfall and potential evapotranspiration.".

Comment 8: Line 278-280: this sentence is not clear, please revise it. Response: Thank you very much for your valuable comment. We corrected "lag time of agricultural drought to meteorological drought" in line 279 as "propagation time from meteorological drought to agricultural drought". "lag time of hydrological drought to meteorological drought" in line 280 were corrected as "lag time from meteorological drought to hydrological drought". Special thanks to you for your good comments.

Please also note the supplement to this comment:
https://nhess.copernicus.org/preprints/nhess-2020-237/nhess-2020-237-AC1-supplement.pdf

---

## Author Comment (AC2) · 15 Jan 2021

Thank you for the reviewers' comments concerning our manuscript entitled "Drought propagation and construction of a comprehensive drought index based on the SWAT-KC': A case study for the Jinta River basin in Northwestern China". Those comments are all valuable and very helpful for revising and improving our paper, as well as the important guiding significance to our researches. We have studied comments carefully and have made correction which we hope meet with approval. The responds to the reviewer's comments are as following:

Major Comments:

Comment 1: Grammar and language. In several places, the grammar and language hinder understanding of the meaning of statements (please see minor comments). This manuscript could be enhanced by carefully proofreading the context. Response: Thanks for the reviewer's suggestion. We reviewed this article based on the language revisions proposed by the reviewer in Minor comments to improve the overall level of the article. The specific modification content is shown in the response to Minor Comments.

Comment 2: Structure. There are three main sections in the results. Are there any connections between them? For example, does the analysis of paragraph time in 3.2.1 help the evaluation or explanation of 3.2.2? In the current form, it seems 3.2.1 and 3.2.2 are independent. This is also reflected in the title "propagation" and "construction". What is the connection? Response: Thanks to the reviewers for your helpful comments. Both drought propagation and the construction of comprehensive drought index are based on the data output by the SWAT hydrological model. Drought propagation is based on the propagation time from meteorological drought to agricultural and hydrological drought, which describes the response relationship between different types of drought. The construction of comprehensive drought index is to combine different types of drought indexes, and include the lag time from meteorological drought to agricultural drought and hydrological drought. It can reflect the drought state when only one or two kinds of drought occur, and can be used to characterize all the characteristics of drought. And those sentences were added in the end of 3.2.1.

Comment 3: Novelty and evaluation. The novelty of the study seems to be the new drought index. A critical question following this is the evaluation of the index. How do we know that this proposed index is reliable? There is some comparison with the individual indicators. Apart from this, could you find some other evidence? For example, what about the public records or reports of historical drought events or losses in the study area? Response: Thanks to the reviewers for helpful comments. According to

[Figure]

the opinions of reviewer, we added information about MAHDI's ability to capture the historical drought events in 3.2.2. We plotted the changes of MAHDI for month scale in No.6 sub-basin, and compared them with the drought months in the study area introduced in "China Meteorological Disaster Dictionary ïĆ§ Gansu Volume" and "Water Resources Bulletin of Gansu Province". We found that MAHDI can capture the historical drought events with records. (1) Line 289, "SSI, SDI, SPEI and applicability analysis of MAHDI" were corrected as "Applicability analysis of MAHDI". (2) Line 290-299, we added "Using the empirical Kendall function to combine the univariate drought indexes, a comprehensive drought index MAHDI that can simultaneously characterize meteorological, agricultural, and hydrological drought was obtained. The monthly change of MAHDI series in the No. 6 sub-watershed from 1986 to 2012 was plotted, as shown in Fig. 5. It can be seen from the figure that 1991, 1999/07∼2000/05, 1994/11∼1995/01, 2009 and 2010/07 were the drought months. According to the " China Meteorological Disaster Dictionary Gansu Volume ", the area was hot and less rainy in 1991, continuous drought occurred in summer and autumn; in 1994-1995, the region suffered from continuous drought in winter and spring; in 1999, the region suffered from severe drought autumn and winter, which were consistent with the drought events described by MAHDI. According to the "Water Resources Bulletin of Gansu Province in 2009", the area had slightly less annual precipitation and higher temperatures. MAHDI also captured the drought in this year. Above all, MAHDI can be used to detect the occurrence and development of drought.". (3) Lines 300-301, the Figure 5 was added.

Figure 5: Monthly-scale MAHDI sequence at the No.6 sub-basin

Minor Comments:

Comment 1: Line 31: please add related references on this. Response: Thank you very much for your valuable comment. Line 31, "(Zhang et al,. 2018)" were added. And "(Zhang et al,. 2018)" were added in line 33. In References, "Zhang, X., Wang, Y., Xiao, W., Yang, R., Wang, Y., and Zhu L.: Responses of net primary productivity of natural vegetation to climate change in the Shi-yang River basin, Chinese Journal of

Ecology, 37, 3110-3118, http://doi.org/10.13292/j.10000-4890.201810.033, 2018. (in Chinese)" were added in line 505-507.

Comment 2: Lines 48-49: This statement can be more concise. Please also check the grammar. Response: Thank you very much for your valuable comment. Line 48-49, the statements of "Drought index is an important role in representing, measuring, and comparing the degree of drought for monitoring, evaluating, and studying the development of drought (drought analysis)." were corrected as "Drought index is an important indicator to characterize and measure the degree of drought, and it can be used to monitor, evaluate and study the development of drought."

Comment 3: Lines 61-64: hard to understand. Please rewrite and make it clear. Response: Thank you very much for your valuable comment. Line 61-64, the statements of "The shortcomings of a univariate drought index gradually emerged with the advancement of drought index research. As drought characteristics are usually interrelated, it is difficult for traditional drought studies that are based on univariate frequency analysis to reflect the complex and extensive characteristics of drought affecting social life. Therefore, it is necessary to develop a comprehensive drought index that integrates multiple variables related to drought." were corrected as "With the deepening of drought research, the insufficiency of univariate drought index has gradually emerged. Because drought characteristics are usually interrelated, traditional drought research based on univariate frequency analysis can hardly reflect the multi-dimensional effects of drought on society. Therefore, it is necessary to develop a comprehensive drought index that takes into account multiple variables related to drought."

Comment 4: Line 68: "ensuring the independence of variables" what do you mean? Response: Thank you very much for your valuable comment. Line 68, ", not only ensuring the independence of variables but also considering" were corrected as "and consider".

Comment 5: Lines 70-71: Please check the grammar. Response: Thank you very

much for your valuable comment. Line 69, ". It is used to develop comprehensive drought indexes" were deleted. ", for example" were corrected as ". For example".

Comment 6: Lines 82-84: This seems to be redundant, as similar statements have been shown in lines 33-34. Response: Thank you very much for your valuable comment. "mid-latitudes of Eurasia and is sensitive to global climate change" were corrected as "climate-sensitive area".

Comment 7: Lines 86-87: "limitation . . . outweigh the advantages" hard to understand. Please justify this or just remove it. Response: Thank you very much for your valuable comment. "However, the limitations of copula function in the construction of a multidimensional drought index outweigh the advantages of the copula." were deleted.

Comment 8: Lines 251-253: "propagation time of agricultural drought to meteorological drought"? How so? (also in lines 264, 275, 279) Do you mean from meteorological drought to agricultural drought? These statements need revisions to make it clear. Response: Thank you very much for your valuable comment. We checked all similar related descriptions and revised them one by one according to the comments of the reviewers. (1) Line 250, "propagation time of agricultural drought to meteorological drought" were corrected as "propagation time from meteorological drought to agricultural drought". (2) Line 259, "propagation time of agricultural drought responding to meteorological drought" were corrected as "propagation time from meteorological drought to agricultural drought". (3) Lines 274-275, "lag time from agricultural drought to meteorological drought" were corrected as "propagation time from meteorological drought to agricultural drought". (4) Line 278, "lag time of agricultural drought to meteorological drought" were corrected as "propagation time from meteorological drought to agricultural drought". (5) Line 279, "lag time of hydrological drought to meteorological drought" were corrected as "lag time from meteorological drought to hydrological drought".

Comment 9: Line 267-268: Could you please elaborate how the "infiltration of soil

water …" could induce the "long lag time"? Response: Thank you very much for your valuable comment. Compared with spring and summer, the evaporation rate of soil water in autumn and winter was slower than that in spring and summer, which prolonged the time when the soil water content reduced to the that of threshold for agricultural drought. This made the agricultural drought lag behind the meteorological drought for a long time. And this reason was added in line 266-268.

Comment 10: Lines 299-301: The statement about "warming and humidification" leading to "increased rainfall and temperature" needs careful justification. Do you have references to support this? Response: Thank you very much for your valuable comment. "The reason for SPEI to exhibit the lowest degree of drought might be due to the warming and humidification of the Shiyang River Basin, which increased rainfall and temperature." were corrected as "SPEI reflected the lowest degree of meteorological drought, which was similar to that described by Thornthwaite aridity index (AI) constructed by Zhang et al. (2017) using rainfall and potential evapotranspiration.".

Comment 11: Lines 308-309: "lack of empirical … with extreme values"? This limitation of the proposed index could be discussed in the conclusion or somewhere else. Response: Thank you very much for your valuable comment. "This might be due to the lack of empirical Kendall function's ability to deal with extreme values." were deleted in 308-309. And "This may be due to the fact that the empirical Kendall function uses nonparametric method to connect three-dimensional sequences, weakening the influence of extremum in the sequence." were added in Conclusion in lines 389-391.

Special thanks to you for your good comments.

Please also note the supplement to this comment:
https://nhess.copernicus.org/preprints/nhess-2020-237/nhess-2020-237-AC2-supplement.pdf
* * *
2020-237, 2020.

[Figure]

**Fig. 1.** Figure 5: Monthly-scale MAHDI sequence at the No.6 sub-basin